# Transcriptome Analysis Revealed Genes Related to γ-Irradiation Induced Emergence Failure in Third-Instar Larvae of *Bactrocera dorsalis*

**DOI:** 10.3390/insects13111017

**Published:** 2022-11-03

**Authors:** Chao Sun, Samina Shabbir, Wenxiang Wang, Yan Gao, Cuicui Ge, Qingsheng Lin

**Affiliations:** 1Guizhou Medical University (GMU), Guiyang 550000, China; 2Plant Protection Research Institute, Guangdong Academy of Agricultural Sciences, Guangzhou 510640, China; 3Guangdong Provincial Key Laboratory of High Technology for Plant Protection, Guangzhou 510640, China; 4Zanyu Technology Group Co., Ltd., Hangzhou 310000, China; 5Zhejiang Gongzheng Testing Centre Co., Ltd., Hangzhou 310000, China

**Keywords:** oriental fruit fly, invasive, quarantine, biomarker, ionizing irradiation

## Abstract

**Simple Summary:**

*Bactrocera dorsalis* (Diptera: Tephritidae) is a severe insect pest of numerous fruits and crops worldwide. Using the Illumina Hiseq 2000 platform, we generated a comprehensive transcriptome of *B. dorsalis* in response to radiation. The adult of *B. dorsalis* could not emerge when third-instar larvae were irradiated with ^60^Co-γ at 116Gy. Most differentially expressed genes (DEGs) in gene ontology (GO) and the Kyoto Encyclopedia of Genes and Genomes (KEEG) in enrichment studies were involved in hemolymph coagulation and digestive processes, such as protein digestion and absorption, while pancreatic secretion may be linked to the response to irradiation. Furthermore, the differential expressions of up/down regulated unigenes were confirmed by quantitative reverse transcription-PCR (qRT-PCR). The downregulation of *sqd* may contribute to the failure of emergence in irradiated samples, while the up-regulation of *ENPEP* and *Ugt* are involved in metabolic pathways, and their up-regulation may be considered the results of the irradiation stress reaction. Overall, the current study is a useful resource for the identification of potentially useful biochemical markers that can be used in quarantine.

**Abstract:**

The oriental fruit fly is a polyphagous and highly invasive economically important pest in the world. We proposed the hypothesis that radiation treatment influence RNA expression in the larvae and leads to emergence failure. Therefore, transcriptome analyses of third-instar larvae of *B. dorsalis* ionizing, irradiated with ^60^Co-γ at 116Gy, were conducted and compared with the controls; a total of 608 DEGs were identified, including 348 up-regulated genes and 260 down-regulated ones. In addition, 130 SNPs in 125 unigenes were identified. For the DEGs, the most significantly enriched GO item was hemolymph coagulation, and some of the enriched pathways were involved in digestive processes. The subsequent validation experiment confirmed the differential expression of six genes, including *sqd*, *ENPEP*, *Jhe*, *mth*, *Notch*, and *Ugt*. Additionally, the 3401:G->A SNP in the *Notch* gene was also successfully validated. According to previous research, this was the first comparative transcriptome study to discover the candidate genes involved in insect molt to pupae. These results not only deepen our understanding of the emerging mechanism of *B. dorsalis* but also provide new insights into the research of biomarkers for quarantine insect treatment with the appropriate dose of radiation.

## 1. Introduction

The oriental fruit fly, *Bactrocera dorsalis* (Diptera: Tephritidae), belongs to the *B. dorsalis* complex and is a severe insect pest of several fruits and vegetables all around the world [1]. It was first detected in Sub-Saharan Africa (SSA) in costal parts of Kenya in 2003 and became widely spread and established successfully across the world [2]. It originates in tropic and sub-tropic regions and is now widespread in China, Southeast Asia, India, and Oceania. Moreover, it has also been found in the Pacific islands, including the Hawaiian Islands, French Polynesia, and Northern Marianas [3]. As an invasive agricultural pest, it causes severe losses to over 250 commercially important plants, including bananas, citrus, star fruit, guava, mango, and eggplants [4]. The female insects oviposit inside the fruit, where the larvae feed until pupation. This usually causes fruit damage and fruit drop [5,6]. Since *B. dorsalis* is polyphagous and highly invasive, quarantine areas for *B. dorsalis* have been declared in many areas [7].

Currently, strict phytosanitary treatments are used to prevent the introduction, and/or spread, of exotic tephritidae pests such as *B. dorsalis*. Chemical fumigants, such as ethylene dibromide (EDB) were first widely used as a pesticide [8]. However, in the late 1980s, EDB was found to be linked to cancer, and in 1984 a ban by the Environmental Protection Agency (EPA), USA, removed EDB from the chemical register for use as an insecticide [9]. Irradiation is an alternative process to chemical fumigation that is easy to apply, quick, and generally safe. Moreover, significant efforts have been made to determine the suitable radiation dose for quarantine treatment. Tephritidae quarantine treatments generally target eggs and larvae because the larvae leave the fruit to pupate in the soil [10]. In addition, the third instar (the last stage of larva) is the most radio-resistant insect stage [11]. Other third-instar tephritid fruit fly larvae have been employed for radiotolerance and substantiative studies, including *B. dorsalis* [12,13,14,15], *Ceratitis capitata* Wiedemann [13,15], *Zeugodacus cucurbitae* Coquillett [13,16], and *Z. tau* Walker [17]. The International Plant Protection Convention (IPPC) is now evaluating prospective standards for the irradiation treatment of *Z. tau*, *B. dorsalis,* and the genus *Anastrepha,* in order to develop international standards as an appendix to ISPM No. 28 [18] and 210, 225, and 250 Gy, where quarantine doses are currently approved for these insects, respectively. Moreover, for tephritid fruit flies and other insects, the United States Department of Agriculture Animal and Plant Health Inspection Service (USDA-APHIS) approved generic radiation treatment doses of 150 Gy and 400 Gy in 2006, respectively [19,20,21], although the adult and pupae of Lepidoptera may require higher doses. In addition, to encourage global usage irradiation of up to 150 Gy would provide quarantine security against all fruit fly species of the family Tephritidae [22]. To prevent the subsequent infection and re-infestation, the fresh commodities would be required to be sterilized with arthropod-proof packaging [16]. Phytosanitary methods, such as temperature, fumigation, and irradiation treatments are currently employed for exporting commodities. The usage of phytosanitary irradiation (PI) treatment has expanded dramatically, with a global fresh goods volume of around 30,000 tons in 2017 [12,23]. Recently, the emergence of next-generation sequencing (NGS) has provided an opportunity for researchers to efficiently obtain massive quantities of sequence data. Since the transcriptome of *B. dorsalis* was built by Illumina, the sequencing and transcriptome analyses of *B. dorsalis* focused on genes involved in reproduction, development, and insecticide resistance have also been performed. The results of previous studies suggested that Illumina transcriptome analysis is a reliable and precise way to study genomic characteristics [24]. Up to now, the comprehensive identification of biochemical markers that can be used in quarantine stations to determine whether the encountering living oriental fruit flies have received an appropriate dose of radiation remains unavailable [25].

In the present study, we present the results from the sequencing and assembly of the transcriptome of two groups (with and without the appropriate dose of radiation with three biological replicates) of the third-instar larvae of *B. dorsalis.* The readings were first assembled through a de novo genome assembly. DEGs were detected and functionally annotated, and the gene expression patterns for the selected DEGs were validated by qRT-PCR. Irradiated-specific single nucleotide polymorphisms (SNPs) or control-specific SNPs shared by all samples in the group were also identified, and some of them were validated by PCR and Sanger sequencing. This transcriptome is a useful resource for the identification of potentially useful biochemical markers that can be used in quarantine. Furthermore, our data will provide insights into the molecular mechanisms putatively related to the irradiation response of *B. dorsalis.*

## 2. Materials and Methods

### 2.1. Insect Rearing and Sample Preparation

The laboratory colony of *B. dorsalis* was from a strain originally collected and reared following the method of Zheng et al. [26] in Guangzhou, Guangdong, China. Adults were fed with water and a standard laboratory diet consisting of yeast powder, honey, sugar, and ascorbic acid. All specimens were kept in an incubator at 27 ± 1 °C, 70 ± 5%, at a relative humidity, and a photoperiod of 12:10(L:D) h. Third-instar larvae were collected and divided into two groups: one group was treated and irradiated with ^60^Co-γ at 116 Gy, according to the Guangzhou Huada Biotechnology Co., Ltd. set manufactured protocol, while the other group was used as controls. Three biological replicates were used for each group: CK (Ck1A, Ck2A, and CK3A) and R (R1A, R2A, and R3A). The Fricke system was employed for both the reference standard and routine dosimetry [27]. This dosimetry system was constructed in accordance with ASTM E1026-13 [27] and ISO/ASTM 51261 [27], and the measured uncertainty value was computed in accordance with ISO/ASTM 51707 [28]. After irradiation, the development from larvae to the pupa of *B. dorsalis* was delayed for about 3 days, and no adults emerged.

### 2.2. RNA Isolation and Sequencing

RNA from 18 flies (an irradiated group and control group with three replicates of each group, including 3 flies per replicate) was extracted using a Trizol-reagent (Invitrogen, Carlsbad, CA, USA), following the manufacturer’s procedures. The quality of the total RNA was verified by measuring its absorbance at 260 nm using a NanoVue UV-Vis spectrophotometer (GE Healthcare Bio-Science, Uppsala, Sweden). The integrity of RNA was verified by 1% agarose gel electrophoresis. Poly (A) mRNA was purified using oligo-dT beads. It was fragmented into short fragments and served as a template to synthesize the first-strand cDNA using a random hexamer-primed reverse transcription, followed by the synthesis of the second-strand cDNA. The cDNA libraries were then prepared following Illumina’s protocols [29] and were sequenced on the Illumina Hiseq 2000 platform with 100 bp paired-end reads in BGI (Beijing Genomics Institute, Shenzhen, Guangdong, China).

### 2.3. De Novo Assembly

Raw data generated by the sequencing were first filtered to remove low-quality reads. Briefly, the following reads were excluded: (1) reads containing adapter sequences; (2) reads with over 5% uncertain nucleotides (Ns); (3) reads with over 10% low-quality bases (Q ≤ 10). The remaining clean reads were used for transcriptome de novo assembly using the Trinity de novo RNAseq assembly pipeline (Inchworm, Chrysalis, and Butterfly) with k-mer sizes set at 25. The output of the Trinity pipeline is a fasta formatted file containing full-length transcripts, erroneous contigs, partial transcript fragments, and non-coding RNA molecules. By using RSEM (RNA-Seq by expectation maximization) software1.2.0 [30], these sequences were then filtered to identify contigs containing full or near full-length transcripts or likely coding regions and isoforms that were represented at a minimum level based on the read abundance. The resulting de novo transcripts (unigenes) were screened against protein databases, such as NR, Swiss-Prot, KEGG, and COG using BLASTX with a cut-off E value of 10^−5^. For each transcript, the best aligning result was used to determine the sequence direction. If different databases gave conflicting results, a priority order of GenBank non-redundant (NR), Swiss-Prot, KEGG, and COG was followed. If a unigene did not map to any of the entries in the above databases, the software ESTScan [31] was introduced to predict its coding regions and sequence direction.

### 2.4. Expression Analysis

Clean reads from each sequencing library were independently mapped to the above transcriptome assembly (only contigs containing full and partial ORFs) by using Soapaligner/soap2 [32]. Mismatches of less than three bases were allowed. Gene expression levels were calculated using the RPKM method [33]. If there was more than one transcript for a given gene, the longest transcript remained to calculate its expression level and coverage. The false discovery rate (FDR) method [34] was used to identify differentially expressed genes (DEGs) between the two groups. Genes with FDR < 0.001 and an absolute value of log2Ratio ≥ 1 (fold change ≥ 2) were selected as DEGs in the DESeq package in R (1.18.0) [35]. Then, the DEGs were further annotated by GO function analysis and KEGG pathway analysis [36].

### 2.5. GO and KEGG Enrichment Analysis

All genes were mapped to the KEGG pathways (http://www.genome.jp/kegg/ (accessed on 1 August 2020)) database and GO database (http://geneontology.org/ (accessed on 1 August 2020)). A hypergeometric distribution test was carried out to identify pathways or GO items overrepresented with DEGs. Multiple testing corrections were performed with the Benjamini-Hochberg procedure [37]. KEEG and GO categories less than *p* < 0.05 were considered highly enriched.

### 2.6. Heterozygous SNP Analysis

The general unigene sequences were used as references for SNP identification. The SNP calling was carried out by using SoapSNP. SoapSNP calculates the likelihood of each genotype based on the alignment results and the corresponding sequencing quality scores. It infers the genotype with the highest posterior probability based on Bayes’ theorem. The SNPs from the two groups were classified into three categories: (1) the common SNPs shared by the irradiated and control samples and the SNPs that were heterozygous in each group; (2) the irradiated-specific SNPs, which were derived only from the irradiated samples and the controls that had the same genotype as the reference; and (3) the control-specific SNPs, which were derived from only the controls and irradiated samples which had the same genotype as the reference. Only irradiated-specific SNPs or control-specific SNPs shared by all the samples in its group remained.

### 2.7. Validation of Gene Expression by qRT-PCR

The differential expressions for the selected genes were validated in two groups (the control and treatment groups), and these genes were picked from the transcriptome data based on their differential expression trends in both groups. In total, we used 18 individuals (from the control and treatment groups) with three biological replicates and three technical replicates. The samples were treated with the same procedure as the samples used for sequencing. RNA was extracted as previously described for sequencing. A total of 2 µg of RNA from each sample was reverse transcribed in a 20 mL reaction system using the PrimerScript TM RT Reagent Kit (Takara Bio Inc., Shiga, Japan). The PCR conditions for both genes were 95 °C for 3 min, followed by 34 cycles of 94 °C for 30 s, 60 °C for 30 s, 72 °C for 30 s, and a final extension at 72 °C for 10 min. The relative fold changes were determined using the 2^−ΔΔCt^ method with the a-tubulin gene (GU269902) of *B. dorsalis* as an internal control. The GraphPad Prism was used to analyze the data (version 7; GraphPad Software, San Diego, CA, USA, https://www.graphpad.com/scientific-software/prism/ (accessed on 27 August 2018)). For mean comparisons, Student’s t-test (*p* > 0.05) was employed. All data were displayed as bar charts, along with the means and standard deviations (Mean ± SD).

### 2.8. SNP Validation

To validate the potential SNP markers, the SNP of the *Notch* gene was further selected and validated into two groups (the control and treatment groups) by PCR and Sanger sequencing [38]. SNP was validated six times, with three biological replicates and three technical replicates, and each replicate had 3 individuals. The samples were treated with the same procedure as the samples used for sequencing. DNA was extracted for the SNP validation experiment. Primers were designed by the Primer Premier 5 software (PREMIER Biosoft International, Palo Alto, CA, USA). PCR amplification in a 50 μL reaction was performed following the conditions: 95 °C for 2 min, 35 cycles of 95 °C for 15 s, 60 °C for 20 s, 72 °C for 30 s, and 72 °C for 2 min. The PCR product purification was carried out using the E.Z.N.A.^®^ Gel Extraction Kit (Omega Bio-tek, Inc., Norcross, GA, USA). Sanger sequencing was performed in both forward and reverse directions on an ABI 3730 DNA Analyzer. The sequence trace files were analyzed manually.

## 3. Results

### 3.1. Sequencing and Transcriptome Assembly

The data generated by Illumina’s Hiseq 2000 sequencing contained a total of 31,987,764,800 bases from 159,938,824 pair-end sequence reads. The average number of reads from the irradiated group and controls were 26,898,794 and 26,414,148, respectively (Appendix A). For each sample, the raw data were firstly assembled into contigs separately (Appendix A), and then the unigenes were combined into the final assembly (Table 1). In total, we generated 50,268 unigenes with the mean length of 781 bp. As shown in Table 1, the unigene size distribution revealed the following: 59.63% (29,975) were between 100 and 500 bp in length; 19.59% (9845) were between 500 and 1000 bp in length; 12.67% (6370) were between 1000 and 2000 bp; and 8.11% (1367) were more than 2000 bp.

### 3.2. Functional Annotation

BLASTX alignments between the predicted unigenes and protein databases, including NR, NT, Swiss-Prot, KEGG, COG, and GO, revealed that 28,358 (56.41%) could be annotated with the known biological functions (Appendix A). The majority of unigenes (79.09%) had a strong homology with *Drosophila* (Figure 1). Among these, 13.28% were best matched to the sequences from *Drosophila melanogaster*, followed by *Drosophila virilis* (11.78%) and *Drosophila willistoni* (11.31%). Other unigenes (20.91%) also had hits with other insect species, such as *Tribolium castaneum* (0.81%) and *Bombyx mori* (0.17%). Compared to species within Diptera, 1.17% of the unigenes matched sequences from *Aedes aegypti*, 0.20% from *Glossina morsitans morsitans*, and 0.14% from *Musca domestica*.

To classify the functions of the unigenes, the COG database was used. As shown in Figure 2, a total of 21,313 unigenes (42.40%) were annotated and divided into 26 specific categories. The general function category, which contained 16.23% of the mapped unigenes (3459), was the top category, followed by the transcription (2017, 9.46%), carbohydrate transport and metabolism (1683, 7.90%), replication, recombination, and repair (1586, 7.44%), and translation, ribosomal structure, and biogenesis (1515, 7.11%). The nuclear structure was the smallest category, which only contained two unigenes (0.0094%). GO analysis divided the unigenes into three ontologies: molecular function, cellular component, and biological process. As shown in Figure 3, 32.65% (16,412 unigenes) of the unigenes were categorized into 59 groups. The cellular process was the group containing the most unigenes.

Pathway mapping analysis was performed based on the KEGG pathway database. As shown in Appendix A, 16,528 unigenes were mapped to 258 pathways. Compared with other, metabolic pathways contained the most unigenes (2421, 14.65%), followed by pathways in cancer (744, 4.5%), focal adhesion (588, 3.56%), and purine metabolism (583, 3.53%).

### 3.3. Differentially Expressed Genes in Response to Irradiation and Pathway Enrichment Analysis of DEGs

To identify the genes that responded to irradiation, differentially expressed sequences between the two groups were identified (Appendix A). There were 608 significantly differentially expressed unigenes detected between the irradiated and control samples, including 348 up-regulated and 260 down-regulated genes. To understand the functions of DEGs, DEGs were mapped to the GO database, and enrichment analysis was performed. A total of 252 DEGs were mapped to the GO database, and 78 GO items were significantly enriched (Table 2). Hemolymph coagulation was the most significantly enriched GO item (*p* = 4.10 × 10^−25^), followed by hemostasis (*p* = 4.83 × 10^−22^), coagulation (*p* = 4.83 × 10^−22^), and the regulation of body fluid levels (*p* = 1.85 × 10^−21^). All of the DEGs were also mapped to the KEGG database and compared with the whole transcriptome background. Among all of the genes with KEGG annotation, a total of 380 DEGs were found between the two groups. A total of 32 pathways were notably enriched (Table 3). The most significant pathway was the renin-angiotensin system (*p* = 5.42 × 10^−10^), followed by protein digestion and absorption (*p* = 3.6 × 10^−8^) and pancreatic secretion (*p* = 3.6 × 10^−8^). Most of the enriched pathways (18/32) were metabolic pathways, such as porphyrin and chlorophyll metabolism, retinol metabolism, steroid hormone biosynthesis, and insect hormone biosynthesis.

### 3.4. Verification of Differentially Expressed Genes

To further evaluate the sequencing results, the expression levels of six genes (eight unigenes) potentially involved in the response to irradiation were analyzed by qRT-PCR. As shown in Table 4 and Figure 4, the validation experiment revealed the same expression tendency as the sequencing data.

### 3.5. SNP Validation

Among the three pair samples for the SNP validation, one pair of the samples failed the PCR experiment, and only another two pairs of the samples were successfully sequenced. Among them, the 3401:G->A SNP in the *Notch* gene was successfully validated (Figure 5), implicating its involvement in response to irradiation.

## 4. Discussion

Irradiation, temperature, and fumigation are currently the most popular phytosanitary treatment techniques used for the disinfestation of postharvest fruit flies in fruits and vegetables [39]. Irradiation has gained popularity as a phytosanitary treatment in the last decade for the eradication of arthropods, ornamentals, and preserved products [40]. Irradiation can cause sub-lethal molecular or biochemical changes, triggering a cascade of physiological changes. In order to eliminate insect pests, gamma radiation is frequently used as a potentially better alternative to toxic fumigants [41].

In this study, with transcriptome sequencing data from two groups (the control and treatment groups with their three biological replicates), we selected molecular markers for the identification of insects that had been treated with gamma irradiation. Previous researchers demonstrated that gamma radiation is environmentally beneficial, with no noticeable environmental adverse effects. Moreover, gamma irradiation prevents the reproduction of several stored grain pest species [42,43].

Our deep sequencing strategy (over 5 G of data for each sample) generated a transcriptome assembly with 50,268 unigenes (a mean length of 781 bp). A total of 20.78% of the unigenes were more than 1000 bp. The previously reported *B. dorsalis* transcriptome, which was also sequenced on the Illumina platform, generated 49,804 unigenes with a mean size of 456 bp, and less than 10% of the unigenes were over 1000 bp. With 454-pyrosequencing, Zheng et al. [44] generated a *B. dorsalis* transcriptome with 48,876 unigenes (mean size of 750 bp), and about 18.4% of the unigenes were over 1000 bp. Therefore, compared with the previously reported *B. dorsalis* transcriptome assembly, we generated more detailed data that facilitated our subsequent analysis. The functional annotation results of the unigenes were generally consistent with the previous transcriptome: the majority of the unigenes had a strong homology with *Drosophila*; the general function category was the top COG category, and the KEGG metabolic pathways contained the most unigenes.

A total of 608 significantly differentially expressed unigenes between the two groups may be attributed to their response to γ irradiation. GO item enrichment analyses revealed that the most significantly enriched GO item was hemolymph coagulation, which is one of the first responses to injury in insects. KEGG pathway analyses of these genes revealed that some of the pathways were involved in digestive processes, such as protein digestion and absorption and pancreatic secretion. This observation is consistent with the well-established irradiation effects on digestive physiology in other insect pests.

The differential expressions of eight unigenes were further confirmed by qRT-PCR. As shown in Table 4 (Figure 4), the squid gene *sqd* was downregulated in irradiated samples while *ENPEP*, *Jhe*, *mth*, *Notch*, and *Ugt* were highly expressed. The downregulation of *sqd* may contribute to the failure of emergence in irradiated samples since the initial organization of the *Drosophila dorsoventral* axis depends on the proteins encoded by *sqd*. Both *ENPEP* and *Ugt* are involved in metabolic pathways, and their up-regulation may be considered the results of the γ irradiation stress reaction. Juvenile hormone esterase (*Jhe*) plays an important role in the regulation of the juvenile hormone (JH). In lepidopteran hemolymph, decreased JH titers are positively correlated with increased JHE abundance [45]. The high expression of *Jhe* here may result in the low titers of JH, which are associated with diapause. For example, Monarch adult diapause is instituted by a deficiency in JH production [46]. Therefore, the upregulation of *Jhe* may contribute to the diapauses of our γ irradiated samples. Individuals with a mutated *mth* displayed an approximately 35 percent increase in their average lifespan and showed significant resistance to oxidative stress, starvation, and heat stress. We also detected a heterozygous SNP in this gene in the γ irradiated samples, which might be caused by the stress response. *Notch* (N) signaling is involved in establishing the correct width of various wing veins [47], and the up-regulation of *Notch* here may also contribute to the failure of emergence in the irradiated samples since the gain of function in *Notch* showed thinner and incomplete wing veins. In addition to the confirmed up-regulation of *Notch*, an SNP in this gene was also validated by the Sanger sequencing. It is highly possible that the confirmed SNP was induced by the irradiation treatment. Since the SNP was validated six times in three biological and three technical replicates of the samples, it could be used as DNA markers in quarantine. Other SNPs identified in this study are also useful sources for further studies on selecting DNA markers in quarantine.

## 5. Conclusions

In conclusion, we have generated a comprehensive transcriptome of the *B. dorsalis* in response to radiation using the Illumina Hiseq 2000 platform. The genes differentially expressed in the controls and irradiated samples were largely identified and functionally annotated. Our results provide useful biochemical markers that could be used in irradiation quarantine treatment. Furthermore, our data provide insights into the molecular mechanisms putatively related to the irradiation response of the third-instar larvae of *B. dorsalis.*

## Figures and Tables

**Figure 1 insects-13-01017-f001:**
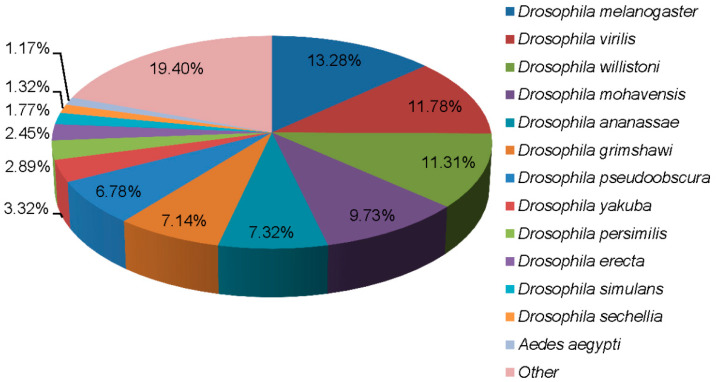
Species distribution of the BLASTX matches (cutoff value E < 10^−5^) of the transcriptome unigenes.

**Figure 2 insects-13-01017-f002:**
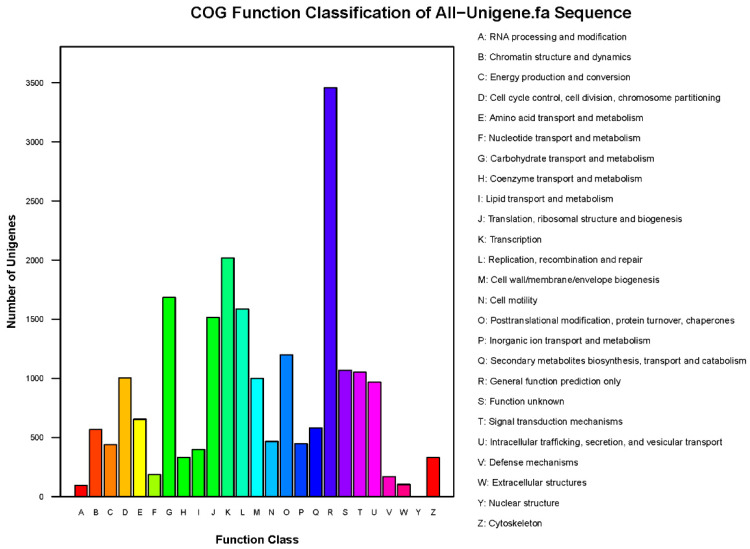
Classification of the clusters of orthologous groups (COG) for the transcriptome.

**Figure 3 insects-13-01017-f003:**
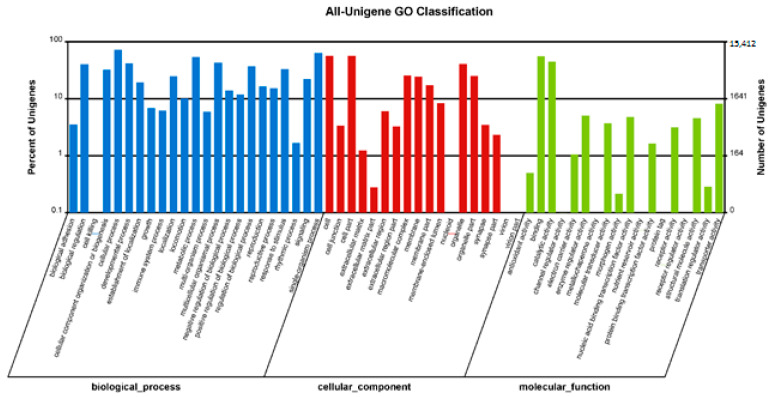
Classification of the gene ontology (GO) for the transcriptome.

**Figure 4 insects-13-01017-f004:**
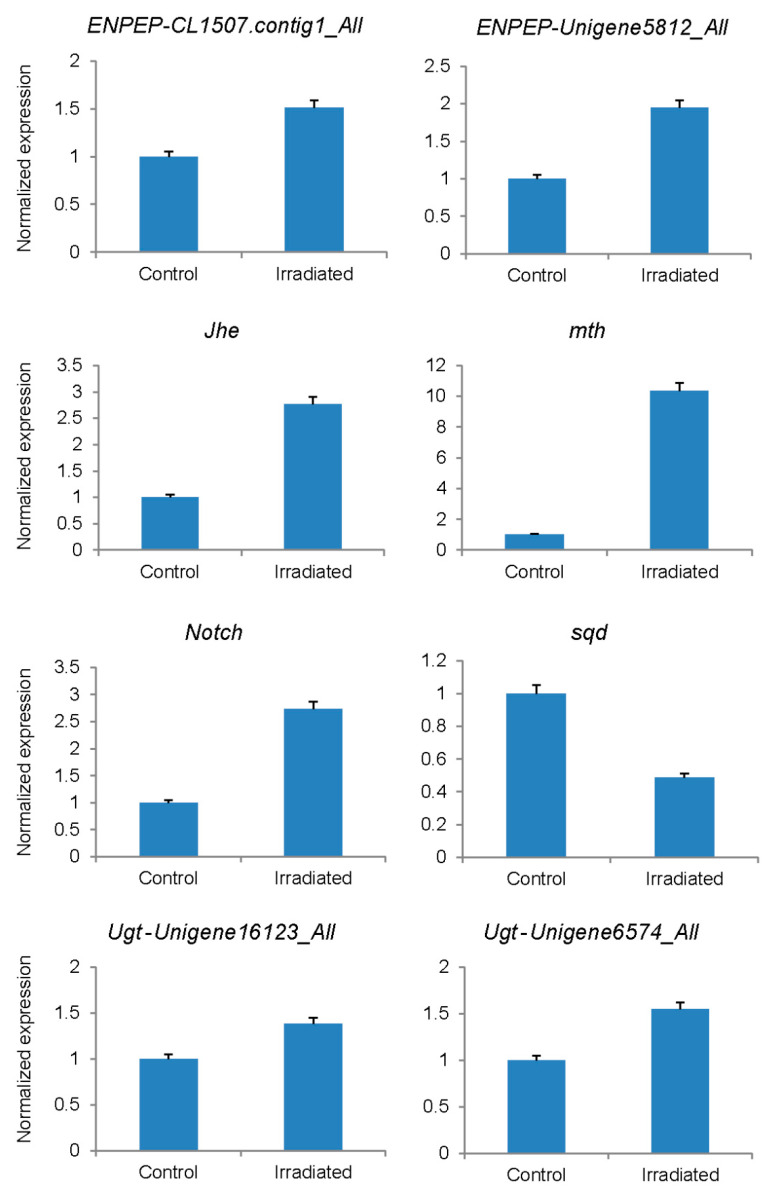
qRT-PCR confirmation of the differentially expressed genes between irradiated samples and controls.

**Figure 5 insects-13-01017-f005:**
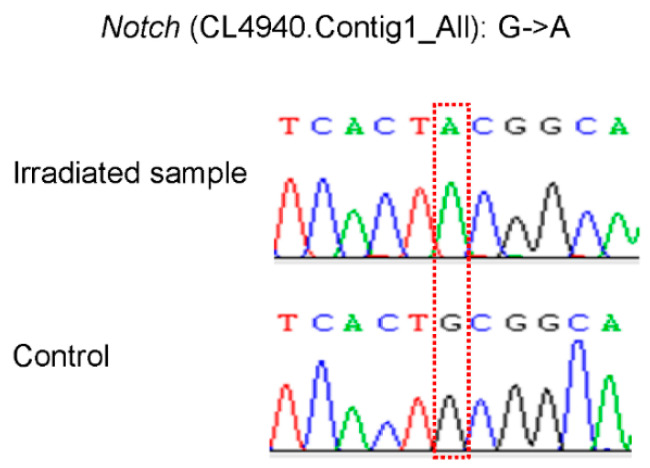
Sanger sequencing results for the SNP identified in the *Notch* gene.

**Table 1 insects-13-01017-t001:** Statistics of assembled unigenes.

Sample	Unigene Counts					N50	Mean Length	All Unigene Counts	Length of All Unigene
	100–500 bp	500–1000 bp	1000–1500 bp	1500–2000 bp	≥2000 bp				
CK1A	42,012	7525	2481	1201	1419	668	466	54,638	25,475,845
CK2A	37,421	6482	2088	986	1116	656	467	48,093	22,476,031
CK3A	38,296	7780	2834	1370	1790	824	525	52,070	27,338,230
R1A	32,563	5640	1683	776	831	608	457	41,493	18,942,224
R2A	34,811	6803	2312	1110	1542	844	530	46,578	24,677,137
R3A	36,509	7816	2864	1630	2248	972	571	51,067	29,140,744
All	29,975	9845	4005	2365	4078	1367	781	50,268	39,256,184

Note: Irradiated samples are shown with the prefix of “R” while control samples are shown with the prefix of “CK”.

**Table 2 insects-13-01017-t002:** Gene ontology term enrichment analyses results for the differentially expressed genes.

Category	Gene Ontology Term	Adjusted *p*-Value
biological process	hemolymph coagulation	4.10× 10^−25^
biological process	hemostasis	4.83 × 10^−22^
biological process	coagulation	4.83 × 10^−22^
biological process	regulation of body fluid levels	1.85 × 10^−21^
molecular function	structural constituent of cuticle	1.30 × 10^−16^
molecular function	structural constituent of chitin-based larval cuticle	5.55 × 10^−16^
cellular component	extracellular region	1.35 × 10^−15^
biological process	wound healing	1.70 × 10^−15^
molecular function	structural constituent of chitin-based cuticle	3.22 × 10^−14^
biological process	response to wounding	1.42 × 10^−12^
biological process	humoral immune response	5.87 × 10^−9^
biological process	defense response	8.80 × 10^−9^
biological process	innate immune response	1.81 × 10^−8^
molecular function	exopeptidase activity	6.87 × 10^−7^
molecular function	catechol oxidase activity	5.89 × 10^−6^
molecular function	L-DOPA monooxygenase activity	5.89 × 10^−6^
molecular function	dopamine monooxygenase activity	5.89 × 10^−6^
biological process	cellular lipid catabolic process	1.60 × 10^−5^
molecular function	peptidase activity	1.73 × 10^−5^
cellular component	intracellular ferritin complex	4.74 × 10^−5^
cellular component	ferritin complex	4.74 × 10^−5^
molecular function	ferrous iron binding	6.24 × 10^−5^
molecular function	peptidase activity, acting on L-amino acid peptides	7.77 × 10^−5^
molecular function	oxidoreductase activity, acting on diphenols and related substances as donors, oxygen as acceptor	7.81 × 10^−5^
biological process	regulation of mitochondrial translation	1.10 × 10^−4^
cellular component	extracellular space	1.60 × 10^−4^
biological process	triglyceride catabolic process	2.40 × 10^−4^
biological process	neutral lipid catabolic process	2.40 × 10^−4^
biological process	acylglycerol catabolic process	2.40 × 10^−4^
biological process	glycerolipid catabolic process	2.40 × 10^−4^
biological process	triglyceride metabolic process	2.70 × 10^−4^
molecular function	aminopeptidase activity	3.50 × 10^−4^
biological process	dopamine metabolic process	5.60 × 10^−4^
biological process	sesquiterpenoid metabolic process	5.70 × 10^−4^
biological process	juvenile hormone metabolic process	5.70 × 10^−4^
biological process	juvenile hormone catabolic process	5.70 × 10^−4^
biological process	isoprenoid catabolic process	5.70 × 10^−4^
biological process	sesquiterpenoid catabolic process	5.70 × 10^−4^
biological process	terpenoid catabolic process	5.70 × 10^−4^
biological process	negative regulation of translational initiation	5.70 × 10^−4^
molecular function	epoxide hydrolase activity	9.00 × 10^−4^
biological process	lipid catabolic process	9.40 × 10^−4^
biological process	immune response	1.08 × 10^−3^
biological process	terpenoid metabolic process	1.71 × 10^−3^
biological process	hormone catabolic process	1.71 × 10^−3^
biological process	catecholamine metabolic process	1.90 × 10^−3^
biological process	catechol-containing compound metabolic process	1.90 × 10^−3^
biological process	phenol-containing compound metabolic process	1.90 × 10^−3^
biological process	diol metabolic process	1.90 × 10^−3^
molecular function	ether hydrolase activity	2.07 × 10^−3^
molecular function	oxidoreductase activity, acting on diphenols and related substances as donors	2.90 × 10^−3^
molecular function	carboxypeptidase activity	3.43 × 10^−3^
biological process	thioester biosynthetic process	3.68 × 10^−3^
biological process	acyl-CoA biosynthetic process	3.68 × 10^−3^
biological process	neutral lipid metabolic process	4.01 × 10^−3^
biological process	acylglycerol metabolic process	4.01 × 10^−3^
biological process	immune system process	4.49 × 10^−3^
biological process	negative regulation of protein metabolic process	4.73 × 10^−3^
molecular function	oxidoreductase activity, acting on paired donors, with incorporation or reduction of molecular oxygen	7.16 × 10^−3^
biological process	monocarboxylic acid metabolic process	7.34 × 10^−3^
molecular function	oxidoreductase activity	9.03 × 10^−3^
biological process	single-organism metabolic process	9.08 × 10^−3^
molecular function	4-aminobutyrate transaminase activity	1.11 × 10^−2^
molecular function	juvenile hormone epoxide hydrolase activity	1.11 × 10^−2^
biological process	lipid metabolic process	1.13 × 10^−2^
molecular function	serine-type peptidase activity	1.27 × 10^−2^
biological process	regulation of mitochondrion organization	1.38 × 10^−2^
biological process	acyl-CoA metabolic process	1.49 × 10^−2^
biological process	thioester metabolic process	1.49 × 10^−2^
molecular function	serine hydrolase activity	1.61 × 10^−2^
molecular function	eukaryotic initiation factor 4E binding	1.82 × 10^−2^
biological process	dsRNA transport	2.16 × 10^−2^
biological process	regulation of hormone metabolic process	2.61 × 10^−2^
molecular function	transferase activity, transferring hexosyl groups	2.82 × 10^−2^
molecular function	electron carrier activity	3.32 × 10^−2^
biological process	fatty acid catabolic process	3.38 × 10^−2^
cellular component	vacuolar proton-transporting V-type ATPase, V1 domain	4.63 × 10^−2^
molecular function	catalytic activity	4.79 × 10^−2^

**Table 3 insects-13-01017-t003:** KEGG pathway enrichment analyses results for the differentially expressed genes.

Pathway ID	Pathway	Adjusted *p* Value	Level 1	Level 2
ko04614	Renin-angiotensin system	5.42 × 10^−10^	Organismal Systems	Endocrine system
ko04974	Protein digestion and absorption	3.60 × 10^−8^	Organismal Systems	Digestive system
ko04972	Pancreatic secretion	3.60 × 10^−8^	Organismal Systems	Digestive system
ko05146	Amoebiasis	3.60 × 10^−8^	Human Diseases	Infectious diseases: Parasitic
ko00860	Porphyrin and chlorophyll metabolism	9.70 × 10^−8^	Metabolism	Metabolism of cofactors and vitamins
ko00830	Retinol metabolism	1.60 × 10^−7^	Metabolism	Metabolism of cofactors and vitamins
ko05110	Vibrio cholerae infection	3.06 × 10^−6^	Human Diseases	Infectious diseases: Bacterial
ko00140	Steroid hormone biosynthesis	5.01 × 10^−6^	Metabolism	Lipid metabolism
ko00514	Other types of O-glycan biosynthesis	5.78 × 10^−5^	Metabolism	Glycan biosynthesis and metabolism
ko00980	Metabolism of xenobiotics by cytochrome P450	5.78 × 10^−5^	Metabolism	Xenobiotics biodegradation and metabolism
ko00982	Drug metabolism—cytochrome P450	6.99 × 10^−5^	Metabolism	Xenobiotics biodegradation and metabolism
ko00740	Riboflavin metabolism	1.09 × 10^−4^	Metabolism	Metabolism of cofactors and vitamins
ko04976	Bile secretion	1.83 × 10^−3^	Organismal Systems	Digestive system
ko00480	Glutathione metabolism	1.83 × 10^−3^	Metabolism	Metabolism of other amino acids
ko00053	Ascorbate and aldarate metabolism	2.27 × 10^−3^	Metabolism	Carbohydrate metabolism
ko05164	Influenza A	2.36 × 10^−3^	Human Diseases	Infectious diseases: Viral
ko05221	Acute myeloid leukemia	2.83 × 10^−3^	Human Diseases	Cancers: Specific types
ko04640	Hematopoietic cell lineage	4.31 × 10^−3^	Organismal Systems	Immune system
ko00130	Ubiquinone and other terpenoid-quinone biosynthesis	8.05 × 10^−3^	Metabolism	Metabolism of cofactors and vitamins
ko00592	alpha-Linolenic acid metabolism	8.05 × 10^−3^	Metabolism	Lipid metabolism
ko00983	Drug metabolism—other enzymes	8.05 × 10^−3^	Metabolism	Xenobiotics biodegradation and metabolism
ko04512	ECM—receptor interaction	1.21 × 10^−2^	Environmental Information Processing	Signaling molecules and interaction
ko00040	Pentose and glucuronate interconversions	1.93 × 10^−2^	Metabolism	Carbohydrate metabolism
ko01100	Metabolic pathways	2.07 × 10^−2^	Metabolism	Global map
ko00350	Tyrosine metabolism	2.31 × 10^−2^	Metabolism	Amino acid metabolism
ko00030	Pentose phosphate pathway	2.48 × 10^−2^	Metabolism	Carbohydrate metabolism
ko04142	Lysosome	2.57 × 10^−2^	Cellular Processes	Transport and catabolism
ko05323	Rheumatoid arthritis	2.57 × 10^−2^	Human Diseases	Immune diseases
ko00981	Insect hormone biosynthesis	2.61 × 10^−2^	Metabolism	Metabolism of terpenoids and polyketides
ko05130	Pathogenic Escherichia coli infection	2.61 × 10^−2^	Human Diseases	Infectious diseases: Bacterial
ko04977	Vitamin digestion and absorption	2.70 × 10^−2^	Organismal Systems	Digestive system
ko00360	Phenylalanine metabolism	4.50 × 10^−2^	Metabolism	Amino acid metabolism

Genes with irradiated-specific SNPs or control-specific SNPs were also considered candidate genes that responded to irradiation. As shown in Appendix A, a total of 130 SNPs in 125 unigenes were identified.

**Table 4 insects-13-01017-t004:** Validation results of the selected differentially expressed genes.

Gene Name	Unigene	Gene Length	Fold Change by Sequencing	*p* Value	Fold by qPCR
*ENPEP*	CL1507.contig1_All	3132	4.02	4.90 × 10^−3^	1.51
*ENPEP*	unigene5812_All	846	8.93	1.29 × 10^−4^	1.95
*Jhe*	CL6471.contig1_All	1713	23.44	3.68 × 10^−2^	2.76
*mth*	CL6547.contig2_All	1044	6.75	4.54 × 10^−4^	10.36
*Notch*	CL4940.contig1_All	9977	5.78	2.27 × 10^−4^	2.73
*sqd*	CL816.contig2_All	687	−4.31	3.14 × 10^−3^	−2.05
*Ugt*	unigene16123_All	1911	4.31	1.07 × 10^−3^	1.38
*Ugt*	unigene6574_All	1007	7.14	2.77 × 10^−4^	1.55

## Data Availability

Data are available in the Appendix A.

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
