# Peer review of "Transcriptome Analysis Revealed Genes Related to γ-Irradiation Induced Emergence Failure in Third-Instar Larvae of Bactrocera dorsalis"

_insects, 2022, doi:10.3390/insects13111017_

Round 1
Reviewer 1 Report
The manuscript described the generation of a comprehensive transcriptome of third-instar larvae of Bactrocera dorsalis in response to radiation. The authors also identified 608 differentially expressed genes (DEGs) and 130 SNPs, which were validated using qRT-PCR and Sanger sequencing. The findings may provide some molecular clues to understand the insect metamorphosis, and are useful for the identification of biomarkers for quarantine insect subject to radiation treatment. My major concern is on the validation of biomarkers. The sampling size, selection standard and experimental operation seems not clearly stated. Bellows are some suggestions.
1. There are some obvious formatting mistakes in both Simple Summary and Abstract. It is described in the “Simple Summary” that “The pupa of B.dorsalis can’t emergence”, while in L106 it is said that “there was no adults emerged”. Please use the right one and keep consistency throughout the manuscript.
2. L32, “overexpressed” would be changed to “up-regulated”.
3. L39, molt to adults or pupae?
4. L108, I would say the sampling size for transcriptome sequencing is small. Therefore, I would suggest the authors clarify the sampling size in the validation of gene expression (L165, 2.7) and SNP (L174, 2.8). The usage of “group” is vague.
5. L230, the title can be improved.
6. L252, 3.4, since 608 DGEs were identified, it is necessary to provide the reason for the six genes selected for qRT-PCR validation.
7. L260, 3.5, as per the results, the experiment seems not to be well conducted. It looks likely that the SNP was only validated in one individual (which was also mentioned in the discussion)? Please provide more evidence and details on this part.
8. It is not well explained in the discussion on how to apply such biomarkers in the quarantine insects, although we may have many choices from the results.
Author Response
Response to Reviewer 1 Comments
The point to point responses to the reviewer’s comments are listed as following:
Point 1: There are some obvious formatting mistakes in both Simple Summary and Abstract. It is described in the “Simple Summary” that “The pupa of B.dorsalis can’t emergence”, while in L106 it is said that “there was no adults emerged”. Please use the right one and keep consistency throughout the manuscript.
Response 1: Thank you very much for the reviewer’s valuable comments because your suggestions are constructive for improving our paper. It was a typing mistake. It has been modified in revised manuscript.
Point 2: 2. L32, “overexpressed” would be changed to “up-regulated”.
Response 2: Thank you very much for the reviewer’s kind suggestion. It has been done.
Point 3: L39, molt to adults or pupae?
Response 3: Molt to pupae. This has been rectified
Point 4: L108, I would say the sampling size for transcriptome sequencing is small. Therefore, I would suggest the authors clarify the sampling size in the validation of gene expression (L165, 2.7) and SNP (L174, 2.8). The usage of “group” is vague.
Response 4: This has been rectified in the revised manuscript. See lines L178-182, 2.7 and L196-197, 2.8. Thanks
Point 5: L230, the title can be improved.
Response 5: Thank you. It has been revised.
Point 6: L252, 3.4, since 608 DGEs were identified, it is necessary to provide the reason for the six genes selected for qRT-PCR validation.
Response 6: Actually, we selected these genes on the base of their higest log2fold change. We added statement on manuscript according to your suggestion, kindly see the lines L180-181, 2.7.
Point 7: L260, 3.5, as per the results, the experiment seems not to be well conducted. It looks likely that the SNP was only validated in one individual (which was also mentioned in the discussion)? Please provide more evidence and details on this part.
Response 7: It has been corrected in revised manuscript. Thank you.
Point 8: It is not well explained in the discussion on how to apply such biomarkers in the quarantine insects, although we may have many choices from the results.
Response 8: It has been modified and improved. Thanks
Reviewer 2 Report
The manuscript “Transcriptome analysis revealed genes related to γ-Irradiation induced emergence failure in third-instar larvae of Oriental 3 Fruit Fly (Bactrocera Dorsalis)” hypothesizes that radiation treatment alters RNA expression in third-instar larvae, leads to failure of emergence. I consider this a relevant study that should be published after significant revisions. One of my primary concerns is the lack of information regarding radiation treatments. The authors do not mention anything about the model of the gamma irradiator, dosimetry system, uncertainty of the dosimetry system, absorbed dose, and DUR in the study. They are evaluating the effect of ionizing radiation on RNA expression of Bactrocera dorsalis third instars, but insufficient information on their radiation treatment is provided in the paper. Without an accurate description of the radiation treatments conducted in the study, the readers will be unable to evaluate the validity of the findings reported in the paper. The introduction needs to be heavily revised to discard inaccurate citations and mention the relevance of their work using valid arguments. I highly recommend the authors consult the International Standards for Phytosanitary Measures (ISPM) from the IPPC, particularly the ISPM 18 and 28, to understand better the phytosanitary certification process associated with phytosanitary irradiation treatments applied commercially around the world. The methods should include detailed information about the radiation treatments and a complete description of their statistical analysis. The results report all relevant information, and data analysis for the qRT-PCR can clarify some of the authors’ considerations. Specific comments were made on the attached pdf revision that may be helpful to the authors.

Round 2
Reviewer 1 Report
I would suggest the authors specify the exact number of the individual(s) used in each part of the study.
